# NON-GAUSSIAN PROCESS REGRESSION

## ABSTRACT

Standard GPs offer a flexible modelling tool for well-behaved processes. However, deviations from Gaussianity are expected to appear in real world datasets, with structural outliers and shocks routinely observed. In these cases GPs can fail to model uncertainty adequately and may over-smooth inferences. Here we extend the GP framework into a new class of time-changed GPs that allow for straightforward modelling of heavy-tailed non-Gaussian behaviours, while retaining a tractable conditional GP structure through an infinite mixture of non-homogeneous GPs representation. The conditional GP structure is obtained by conditioning the observations on a latent transformed input space and the random evolution of the latent transformation is modelled using a Lévy process which allows Bayesian inference in both the posterior predictive density and the latent transformation function. We present Markov chain Monte Carlo inference procedures for this model and demonstrate the potential benefits compared to a standard GP.

## 1 INTRODUCTION

Gaussian processes (GPs) are stochastic processes which are widely used in nonparametric regression and classification problems to represent probability distributions over functions (Rasmussen & Williams (2006)). They allow Bayesian inference in a space of functions such that consistent uncertainty measures over predictions are obtained rather than only point estimates. In its simplest form a GP defines a distribution over functions through its particular mean and covariance (kernel) functions which determine the smoothness, stationarity and periodicity of a random realisation in the function space. As a prior distribution in Bayesian inference, using a zero mean GP reflects the lack of information in the values and trend of the function. In this case the covariance function, which defines the similarity between any two points in the input space, fully characterises the properties of the random function space.

The design of kernel functions that are able to represent a wide range of characteristics and make consistent generalisations is a fundamental area of research. Some recent work in this area include modelling the kernel via spectral densities that are scale-location mixtures of Gaussians (Wilson & Adams (2013)), and similarly using Lévy process priors over adaptive basis expansions for the spectral density (Jang et al. (2017)). Furthermore, extensions to the standard GP model can be made by directly modelling the covariance matrix as a stochastic process (Wilson & Ghahramani (2011)), assuming heteroscedastic noise on the observations and carrying out variational inference (Lázaro-Gredilla & Titsias (2011)), or learning nonlinear transformations of the observations such that the latent transformed observations are modelled well by a GP (Snelson et al. (2003); Lázaro-Gredilla (2012)). Nonstationarity in the measurement process can be expressed as a product of multiple GPs (Adams & Stegle (2008)) and heavy-tailed observations may be modelled through the Student-t process (Shah et al. (2014)). Particularly relevant extensions of GP models are presented in (Rasmussen & Ghahramani (2001)) where the input space is locally modelled by separate GPs, and string GPs (Samo & Roberts (2016)) introduce link functions between local GPs such that the global process is still a GP and provides efficient inference methods on large data sets. In (Schmidt & O'Hagan (2003); Snoek et al. (2014)) a latent space is defined between the inputs and observations through a separate GP and a class of bounded functions in $[0, 1]$, respectively.

By designing expressive covariance functions or stacking multiple GPs in structured arrangements, the GP framework produces accurate predictive models in numerous application domains. However, these models are limited by their Gaussianity assumption such that the local patterns learned

through these models are highly dependent on particular observations instead of learning the overall dynamics of the data generating system. A more natural and interpretable way to define complex relationships may be to assume that the underlying random function is non-Gaussian which yields more sparse representations (Unser & Tafti (2014)) as discussed in Section 4.

In this work, we present a novel approach to modelling non-Gaussian dynamics by constructing a non-Gaussian process (NGP) such that the observations form a conditional GP that is conditioned on a latent input transformation function that is separately modelled as a Lévy process. Building on the definition of a stationary kernel, the latent layer between the input and output spaces represent the random distances between any two points on an input space. In order to define the distribution of random distances without referring to a specific origin, and in order to maintain monotonicity of the input space transformation, the latent space of transformation functions is modelled by a special class of Lévy process called a subordinator that is non-negative and non-decreasing. Such a process is characterised by the distribution of its stationary and independent increments which as a result defines a probability distribution over the distance between any two input values. Making random monotonic transformations of input values allow the kernel to adapt to the local characteristics of an input space or in other words to the varying rate of change in the observations and the learned subordinator provides uncertainty estimates over the variation of the observed process everywhere on its domain. In this paper we focus principally upon 1-dimensional GPs for the sake of brevity, but we emphasise that our approach can be readily extended to multiple dimensions, as described throughout the text and illustrated in the experimental results.

NGPs are related to continuous-time stochastic volatility models studied in the mathematical finance literature to model the behaviour of a stochastic process which has a randomly distributed variance (Ghysels et al. (1996)). The time-change operation defined for continuous-time stochastic processes is a standard approach to building stochastic volatility models. A common example is the time-changed Brownian motion where the time-change is chosen to be a subordinator and the time-changed motion produces a Lévy process (Veraart & Winkel (2010)). In such a model, the process is conditionally a Brownian motion i.e. the integral of a white-noise GP. Similarly, our construction of a stationary NGP follows a GP conditioned on the latent values of a subordinator, thus it is a time-changed GP. Particular non-Gaussian behaviour can be expressed through the characterisation of a subordinator, examples include the stable law, normal-tempered stable, and generalised hyperbolic (including Student-t) processes. Hence, NGPs provide a flexible and expressive probabilistic framework for nonparametric learning of functions.

In Section 2 we briefly review the GP regression framework, introduce the time-change operation and define NGPs. An inference method for NGP regression is presented in Section 3 following an introduction to shot-noise simulation methods for Lévy processes. In Section 4, we present the results of applying NGP regression on representative synthetically generated non-Gaussian data sets to visually highlight their dynamics and compare the results to alternative GPs. Furthermore, a multidimensional example using a data set available in TensorFlow is presented.

## 2 MODELS

In this section, we briefly present the standard GP regression framework and introduce the time-change operation which results in a non-Gaussian process (NGP). The series representation of a Lévy process (Rosiński (2001)) reviewed in Section 2.2 is central to the inference methodology studied in Section 3.

### 2.1 GAUSSIAN PROCESS REGRESSION

A stochastic process $\{f(x) \in \mathbb{R}; x \in \mathcal{X}\}$ is defined by the probability distribution of all possible finite subsets of its values, where $\mathcal{X} \in \mathbb{R}^d$ is a $d$-dimensional input space. In the case of GPs, for any finite set of inputs $\{x_i\}_{i=1}^n$ the corresponding values of the function $\{f(x_i)\}_{i=1}^n$ has a multivariate Gaussian distribution (MacKay (2003)) characterised by its mean $m(x) = \mathbb{E}[f(x)]$ and covariance kernel functions $K(x', x) = \text{Cov}(f(x'), f(x))$ where $x', x \in \mathcal{X}$. Given a set of inputs $\{x_i\}$ the mean function forms a vector $\mathbf{m}$ and the kernel function forms a positive-definite covariance matrix $\mathbf{\Sigma}$. The resulting multivariate Gaussian distribution can be extended to any input $x^* \in \mathcal{X}$ follow-

ing the Kolmogorov extension theorem (Øksendal (2014)) which produces an interpretation of the stochastic process as a random function $f$ such that $p(f) \sim \mathcal{GP}(m(x), K(x', x))$.

In the standard GP regression setting, it is assumed that noisy observations of a function $f(x)$ are made such that $y_{1:n} = f(x_{1:n}) + \epsilon_{1:n}$ where $\epsilon_{1:n} \sim \mathcal{N}(0, \mathbf{\Omega})$ are a sequence of independent and identically distributed Gaussian noise. Following a Bayesian approach, a prior distribution on the function space is defined such that $p(f) \sim \mathcal{GP}(0, K(x', x))$ where any marginal $f_i = f(x_i)$ has a Gaussian distribution. In general referring to any particular marginal of $f$ is not necessary since a GP is defined for all points in $\mathcal{X}$ and the finite distribution is understood from the context. Since the likelihood $p(y_{1:n}|f)$ is a product of Gaussians, the posterior distribution over the function space can be analytically found to be a GP with a particular mean $\bar{m}(\cdot)$ and kernel function $\bar{K}(\cdot, \cdot)$ where both functions are defined for any finite set of inputs as shown in (Rasmussen & Williams (2006)). The posterior GP is denoted as $p(f|y_{1:n}) \sim \mathcal{GP}(\bar{m}, \bar{K})$.

## 2.2 TIME-CHANGE

In this section, the classical time-change operation is introduced in one dimension of time and the operation is generalised to multidimensional input spaces using subordinated Gaussian fields (Dobrushin (1979); Merkle & Barth (2022a;b)).

Let $\{g(t)\}_{t \geq 0}$ be an isotropic stochastic process which has uniformly distributed variance. The operational time $t$ can be interpreted as a linear representation of change such that the derivative $dt$ is proportional to a deterministic constant. Hence the variance of $g(t)$ scales proportionally to time intervals. Random evolution in variance may be obtained by considering a representation of change that is random and nonlinear. Hence define a non-negative, non-decreasing stochastic process $\{W(t)\}_{t \geq 0}$ such that it randomly maps time instances while preserving their order, therefore changing the time. A time-changed stochastic process $\{f(t)\}_{t \geq 0}$ is then defined as $f(t) = g(W(t))$ where the evolution of $f$ is governed by $dW(t)$ instead of $dt$. In other words, the change in $f$ will have variance proportional to $W(t) - W(s)$, instead of $t - s$ where $t > s$. Assuming that $g(t)$ is Gaussian, this operation enables large deviations from Gaussian behaviour to occur when $dW(t)$ is large, while retaining a conditionally Gaussian form.

The random evolution of $W(t)$ can be modelled as a subordinator that take values in $[0, \infty)$ such that it has independent and stationary increments with no fixed discontinuities (Feller (1966); Bertoin (1997)). Thus, a subordinator increases non-linearly with a certain statistical distribution defined by the random number of discontinuities and their random magnitudes. A Lévy process $W(t)$ in $[0, \infty)$ having no drift or Brownian motion is defined through its characteristic function $\mathbb{E}\left[\exp(iuW(t))\right] = \exp\left(t\left[\int_{(0,\infty)}(e^{iuw} - 1)Q(dw)\right]\right)$ (Kallenberg (2002), Corollary 15.8) where $Q$ is a Lévy measure that satisfies $\int_{(0,\infty)}(1 \wedge x)Q(dx) < \infty$ (Bertoin (1997), p.72). The Lévy measure $Q$ is defined on the random magnitudes of discontinuities, called jumps, and denotes the expected number of jumps per unit time whose magnitudes belong to some subset of the jump space (Tankov, P. and Cont, R. (2015)).

By the Lévy-Itô decomposition, a pure jump Lévy process (i.e. containing no Brownian motion) may be expressed using a stochastic integral as

$$W(t) = \int_{(0,\infty)} wN([0, t], dw) \tag{1}$$

where $N$ is a bivariate point process having mean measure $Leb \times Q$ on $[0, T] \times (0, \infty)$ which can be conveniently expressed using a Poisson random measure as

$$N = \sum_{i=1}^{\infty} \delta_{V_i, M_i} \tag{2}$$

where $\{V_i \in [0, T]\}$ are i.i.d. uniform random variables which give the times of arrival of jumps, $\{M_i\}$ are the sizes of the jumps and $\delta_{V_i, M_i}$ is Dirac measure centered at time $V_i$ and jump size $M_i$. Substituting $N$ into Eq. (1) leads to a representation of a Lévy jump process as an infinite series

$$W(t) = \sum_{i=1}^{\infty} M_i \mathcal{I}_{V_i \leq t} \quad a.s. \tag{3}$$

The almost sure convergence of this series to $\{W(t)\}$ is proved in (Rosiński (2001)). Therefore, by sampling pairs of jump times and sizes $\{V_i, M_i\}$, a realisation of a Lévy process $W(t)$ may be obtained.

The standard formulation of the time-change operation on $[0, \infty)$ can be extended to $d$-dimensional input spaces $\mathcal{X}$ by considering the Poisson random measure representation $N$ of a Lévy process. A homogeneous Poisson process expressing the jump times can be generalised to any number of dimensions where we define arbitrary inputs $x', x \in \mathbb{R}^d$ (Kingman, J.F.C. (1992)). The independence properties of a Lévy measure allow the definition on the unit time interval to be extended to unit $d$-dimensional volumes by appropriately scaling the rate of the process (Wolpert & Ickstadt (1998b)). For multidimensional input transformations a subordination field on $\mathbb{R}^d$ is a $d$-dimensional stochastic process such that each of its dimensions is a subordinator. Thus the $i$-th dimension of an input vector $x^{(i)}$ is mapped to $W^{(i)}(x^{(i)})$ where $W^{(i)}$ denotes the subordinator on $i$. Therefore a distance $d(x', x)$ can be randomly transformed as $d(W(x'), W(x))$. Hence, the choice of a Lévy measure characterise the distribution of the random distances over the input space. The notation introduced for the multidimensional treatment of subordination is omitted for brevity in the following sections as it is straightforward to extend the model into multidimensional input spaces.

### 2.3 NON-GAUSSIAN PROCESSES

A non-Gaussian process (NGP) prior on functions can be obtained by randomly transforming the inputs using a subordinator and carrying out GP regression on the transformed input space. The resulting posterior distribution follows a non-Gaussian stochastic process. Given a set of input-output pairs $\{x_i, y_i\}$ consider a latent input transformation such that $x_i$ is mapped to $W(x_i)$ where $\{W(x); x \in \mathcal{X}\}$ is a subordinator. The associated prior on the transformation function is then defined as $p(W)$ and the conditional prior over $f$ is $p(f|W) \sim \mathcal{GP}(m_W(x), K_W(x', x))$ where $m_W(x) = m(W(x))$, $K_W(x', x) = K(W(x'), W(x)) = K(|W(x') - W(x)|)$ and $K(\cdot, \cdot)$ is a stationary kernel function e.g. squared exponential or Matérn. The joint distribution over the product space of $f$ and $W$, $p(f, W|y_{1:n})$ characterises the NGP prior.

The conditional GP structure of a NGP induces a posterior mean $\bar{m}_W(\cdot)$ and kernel function $\bar{K}_W(\cdot, \cdot)$ that can be evaluated analytically, i.e. $p(f|y_{1:n}, W) \sim \mathcal{GP}(\bar{m}_W, \bar{K}_W)$. The conditional likelihood $p(y_{1:n}|W)$ is of particular interest in this framework since it is a measure of how well the data is represented by the model given a random transformation and it can also be evaluated analytically.

The NGP posterior distribution over the function space is found as

$$p(f|y_{1:n}) = \int p(f|y_{1:n}, W) p(W|y_{1:n}) dW$$

where $p(W|y_{1:n})$ is the posterior distribution of the subordinator process. Inferring $p(W|y_{1:n})$ and hence $p(f|y_{1:n})$ is analytically intractable, however using approximate inference methods allow for straightforward extensions of the model and fully Bayesian inference as discussed in the following.

## 3 SAMPLING AND INFERENCE

In this section, we review shot-noise simulation methods for simulating Lévy processes based on series representations (Rosiński (2001)). We describe a novel Metropolis-Hastings-within-Gibbs (MH-in-Gibbs) algorithm (Hastings (1970); Chib & Greenberg (1995)) to obtain samples from the posterior distribution of a subordinator and estimate a non-Gaussian process posterior $p(f|y_{1:n})$.

## 3.1 SHOT-NOISE SIMULATION METHODS

The jump magnitudes $\{M_i\}_{i=1}^{\infty}$ shown in Eq. (3) of a Lévy process cannot be directly simulated because there may be an infinite number of jumps in any finite interval. One way to obtain approximate samples from such an infinite sequence is to consider ordering the jump magnitudes by size and simulating large jumps while ignoring or approximating the residual error as discussed in (Ferguson & Klass (1972); Rosiński (2001); Wolpert & Ickstadt (1998a;b); Godsill & Kındap (2021)). Once the ordered jump sizes have been obtained, the corresponding jump positions $\{V_i\}_{i=1}^{\infty}$ may be simulated independently from a uniform distribution on $(x_{lb}, x_{ub})$ where $x_{lb}, x_{ub}$ are some lower and upper bounds, or sequentially in space from a homogeneous Poisson process if preferred.

Consider a bivariate point process $N'$ that has the same form as Eq. (2) where the jump magnitudes $M_i$ are expressed as the output of a function $h(\Gamma_i)$ where $\{\Gamma_i\}_{i=1}^{\infty}$ are the epochs of a unit rate Poisson process, i.e. the cumulative sum of exponential random variables with unit rate, independent of $\{V_i\}_{i=1}^{\infty}$. Similar to the standard inverse CDF method for random variate generation, the upper tail mass of a Lévy measure $Q^+(x) = Q([x, \infty))$ can be inverted to produce jump magnitudes of a subordinator by passing epochs of a homogeneous Poisson process through the inverse Lévy measure $Q^{+^{-1}}(\cdot)$. The corresponding function $h(\cdot) = Q^{+^{-1}}(\cdot)$ is non-increasing thus $\{h(\Gamma_i)\}$ is an ordered sequence representing random jump sizes. Note that the epochs of a homogeneous Poisson process are analogous to uniformly distributed random variables in $(0, \infty)$ and the mapping theorem states that the resulting points $\{V_i, h(\Gamma_i)\}$ form a Poisson point process $N' = \sum_{i=1}^{\infty} \delta_{V_i, h(\Gamma_i)}$ on $(x_{lb}, x_{ub}) \times (0, \infty)$ (Kingman, J.F.C. (1992)). $N'$ converges almost surely to $N$ as the $\{\Gamma_i\}$ sequence is simulated indefinitely (Rosiński (2001)) and approximations of the point process may be obtained through finite samples.

The explicit evaluation of the inverse tail measure $Q^{+^{-1}}(\cdot)$ in general is not possible. The Lévy measures considered in this paper possess a density function denoted as $Q(x)$ such that $Q(dx) = Q(x)dx$. The approach taken in this work is to simulate from a tractable dominating point process $N_0$ having Lévy measure $Q_0$ such that $Q_0(dx)/Q(dx) \geq 1, \forall x \in (0, \infty)$ for which $h(\cdot)$ can be explicitly evaluated. The resulting jump magnitudes belonging to $N_0$ are then thinned with probability $Q(x)/Q_0(x)$ as in (Lewis & Shedler (1979)) to obtain the desired approximate jump magnitudes $\{M_i\}$ of a subordinator.

As a motivating example in this paper, we consider tempered stable (TS) processes which are commonly used in mathematical finance to model stochastic volatility (Carr et al. (2003)). We note that our methodology is equally applicable to other subordinator processes for which shot noise simulation methods can be applied (Godsill & Kındap (2021); Godsill et al. (2019)). A TS process exhibits both $\alpha$-stable and Gaussian trends depending on the distance it travels. For short distances the stable characteristics prevail and the TS process produces larger jumps compared to a Gaussian process. For longer distances the tempering causes a TS process to produce Gaussian trends (Rosiński (2007)). Thus, a TS process is a natural extension to Gaussian processes.

The Lévy density for the subordinator TS process is defined as (Shephard & Barndorff-Nielsen (2001); Rosiński (2007))

$$Q(x) = Cx^{-1-\alpha}e^{-\beta x}, \qquad x > 0 \qquad (4)$$

where $\alpha \in (0, 1)$ is the tail parameter and $\beta$ is the tempering parameter. The corresponding tail probability may be calculated in terms of gamma functions but it cannot be analytically inverted and numerical approximations are needed (Imai & Kawai (2011)). Instead, we adopt a thinning approach where the Lévy density is factorised into a $\alpha$-stable subordinator process with Lévy density $Q_0(x) = Cx^{-1-\alpha}$ (Samorodnitsky & Taqqu (1994); Godsill et al. (2019)) and a tempering function $e^{-\beta x}$. The tail mass of a stable process can be found to be $Q_0^+(x) = \frac{C}{\alpha}x^{-\alpha}$ and inverting this function produces the simulation function $h(\gamma) = \left(\frac{\alpha\gamma}{C}\right)^{-1/\alpha}$. Given points $x_i$ from a stable point process with density $Q_0(x)$, individually selecting (thinning) points with probability $e^{-\beta x_i}$ results in a tempered stable process. The associated sampling algorithm is shown in Alg. (1) for reference.

Alg. (1) generates the jumps that correspond to a TS process in $(0, 1)$. Since the jumps of a Lévy process are independent and stationary it is straightforward to adjust the interval. For instance, setting the rate of the underlying Poisson process produced in the second stage of Alg. (1) to the

---

**Algorithm 1** Generation of the jumps of a tempered stable process with Lévy density $Q_{TS}(x) = Cx^{-1-\alpha}e^{-\beta x}$ where $\alpha$ is the tail parameter and $\beta$ is the tempering parameter.

---

1. Assign $N_{TS} = \emptyset$,
2. Generate the epochs of a unit rate Poisson process, $\{\Gamma_i; \ i = 1, 2, 3...\}$,
3. For $i = 1, 2, 3...$,
   - Compute $x_i = \left(\frac{\alpha \Gamma_i}{C}\right)^{-1/\alpha}$,
   - With probability $e^{-\beta x_i}$, accept $x_i$ and assign $N_{TS} = N_{TS} \cup x_i$.

---

length of the interval $(x_{lb}, x_{ub})$ produces the associated TS process. Similarly, for $d$-dimensional input spaces the jumps on a $n$-dimensional hypercube can be simulated by setting the rate to the associated volume (Wolpert & Ickstadt (1998b)).

## 3.2 Approximate inference

Since a stochastic process is defined as an infinite collection of random variables, designing direct sampling methods from the posterior $p(W|y_{1:n})$ based on batch Monte Carlo methods is a difficult task. Instead a more appropriate approach to high dimensional problems is to use a Gibbs sampler which approximates samples from a multivariate probability distribution or in this case a stochastic process.

A Gibbs sampler approximating samples from $p(W|y_{1:n})$ can be implemented by simulating the associated bivariate random points that define the jump size and position on small disjoint intervals $\tau = (x_j, x_l)$ conditioned on the previous sample points in $-\tau = \mathcal{X} \setminus (x_j, x_l)$ and observations. Progressively simulating these points such that the whole input space is covered leads to approximate samples from the target distribution. While Gibbs sampling reduces the complexity of sampling a stochastic process for each small interval, direct sampling from the conditional posterior for each interval is still intractable in general. Thus for each interval a Metropolis-Hastings algorithm is used yielding a MH-within-Gibbs sampling algorithm (Chib & Greenberg (1995)). The proposal density for the MCMC sampling procedure is $p(W_\tau|W_{-\tau})$ which produces new bivariate points (jump sizes and times) on some interval $\tau$ conditioned on all points in $-\tau$ as described in Alg. (2).

---

**Algorithm 2** Simulating sample paths from the proposal density $p(W_\tau|W_{-\tau})$.

---

Given a random length set $N_W = \{V_i^{(k)}, M_i^{(k)}\}$ and an interval $(x_j, x_l) \in \mathcal{X}$,

1. Simulate $\{V_i^{(\prime)}, M_i^{(\prime)}\}$ with rate $|x_j - x_l|$ using Algorithm 1,
2. Remove all points $\{V_i^{(k)}, M_i^{(k)}\}$ from $N_W$ such that $x_j < V_i^{(k)} < x_l$ and add $\{V_i^{(\prime)}, M_i^{(\prime)}\}$, $N_W = N_W \cup \{V_i^{(\prime)}, M_i^{(\prime)}\}$,
3. Substitute the points of $N_W$ into Eq. 3 to obtain the proposed sample path $W^{(\prime)}$.

---

For each realisation of the subordinator $W^{(k)}$, the conditional likelihood $p(y_{1:n}|W^{(k)})$ may be used as a weight in a Markov chain Monte Carlo sampler since it is proportional to the posterior distribution $p(W|y_{1:n})$ and we are proposing from $p(W_\tau|W_{-\tau})$. As discussed in Section 2.3 the conditional likelihood $p(y_{1:n}|W^{(k)})$ may be analytically found given the values of $W^{(k)}$. Then given a sample $W^{(k)}$ and proposal $W^{(\prime)}$, the acceptance probability for the proposal is

$$\alpha(W^{(\prime)}, W^{(k)}) = \min\left(1, \frac{p(y_{1:n}|W^{(\prime)})}{p(y_{1:n}|W^{(k)})}\right) \tag{5}$$

---

**Algorithm 3** MH-within-Gibbs sampler for $p(W|y_{1:n})$.

1. Initialise $W^{(0)}$ by simulating $\{V_i, M_i\}$ from the associated bivariate point process using Alg. 1,

2. Analytically evaluate $\bar{m}_{W^{(0)}}$, $\bar{K}_{W^{(0)}}$ which define the conditional GP posterior $p(f|y_{1:n}, W^{(0)})$, and the conditional likelihood $p(y_{1:n}|W^{(0)})$,

3. For $N$ times, iterate over $\tau_j \in \mathcal{X}$ where $\bigcup_{j=1}^{J} \tau_j = \mathcal{X}$,

   (a) Using $\tau_j$ and the points $\{V_i^{(k)}, M_i^{(k)}\}$ associated with $W^{(k)}$, sample a proposed sample path $W^{(\prime)}$ using Alg. 2,

   (b) Evaluate $\bar{m}_{W^{(\prime)}}$, $\bar{K}_{W^{(\prime)}}$ and $p(y_{1:n}|W^{(\prime)})$,

   (c) With probability $\alpha(W^{(\prime)}, W^{(k)})$ the proposal is accepted and $W^{(k+1)} = W^{(\prime)}$, otherwise reject and set $W^{(k+1)} = W^{(k)}$.

---

The MH-within-Gibbs sampling procedure is described in Alg. (3). The resulting samples $\{W^{(k)}\}$ are individually associated with conditional GP posterior functions $p(f|y_{1:n}, W^{(k)})$ that are completely defined through their mean $\bar{m}_{W^{(k)}}$ and covariance $\bar{K}_{W^{(k)}}$ functions. Such a collection forms a Gaussian mixture distribution and the mean and covariance of the corresponding mixture density can be obtained as

$$\mathbb{E}_{f|\mathbf{y}}[f] = \frac{1}{N} \sum_{k=1}^{N} \bar{m}_{W^{(k)}} = m_{f|\mathbf{y}} \tag{6}$$

and

$$\mathrm{Cov}_{f|\mathbf{y}}(f) = \frac{1}{N} \sum_{k=1}^{N} \left[ \bar{K}_{W^{(k)}} + (\bar{m}_{W^{(k)}} - m_{f|\mathbf{y}})(\bar{m}_{W^{(k)}} - m_{f|\mathbf{y}})^T \right] \tag{7}$$

where $N$ is the number of samples and $\mathbb{E}_{f|\mathbf{y}}[f]$, $\mathrm{Cov}_{f|\mathbf{y}}(f)$ define the posterior mean and covariance of the random function $f$. It is straightforward to obtain the corresponding predictive density $p(y^*|y_{1:N})$ by adding the observation noise matrix $\mathbf{\Omega}$ to each covariance matrix sample $\bar{K}_{W^{(k)}}$. Using a constant noise matrix $\mathbf{\Omega}$ corresponds to the assumption that the observation likelihood model is Gaussian (Rasmussen & Williams (2006)). This assumption can be relaxed by sampling a noise matrix $\mathbf{\Omega}^{(k)}$ for each individual sample to consider non-Gaussian likelihood models such as scale mixture of normals which includes the Student-t and Laplace distributions (Barndorff-Nielsen et al. (1982)). This results in doubly non-Gaussian behaviour which is highly expressive while retaining interpretation of individual components of the behaviour.

Following a similar approach the hyperparameters $C$, $\alpha$ and $\beta$ of the subordinator process may be included in the sampling procedure by considering an appropriate prior distribution over their values. Hence these parameters may be marginalised out using the Monte Carlo procedure, which leaves the same number of kernel parameters that define a standard GP. This approach works successfully and will be reported in a future publication. Furthermore, a nonparametric kernel may be included in this framework by considering a prior distribution on stationary kernel functions and sampling a kernel function for each proposed sample. Some examples of nonparametric kernel design can be found in (Wilson & Adams (2013); Tobar et al. (2015); Bruinsma et al. (2022)).

A straightforward extension of Alg. (3) to multidimensional input spaces can be achieved by assuming that individual subordinator dimensions $x^{(i)}$ are independent *a priori*. The simulation steps defined by Alg. (1) and (2) can be independently applied to each dimension and the other steps remain unchanged, replacing step 3. (b) with the multidimensional GP likelihood.

## 4 EXPERIMENTAL RESULTS AND DISCUSSION

In this section, we present the results of applying NGP regression to non-Gaussian data sets and compare the results with alternative standard GP regression settings. In order to emphasise the differences in non-Gaussian and Gaussian processes, we first use a synthetically generated data set

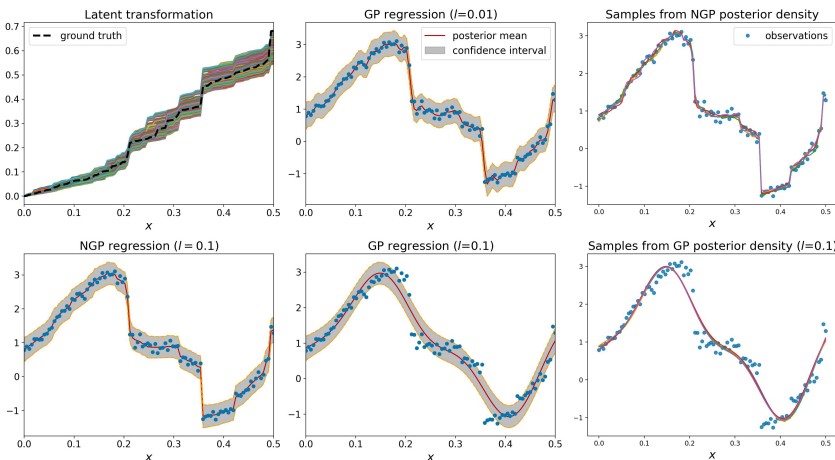

Figure 1: Regression analysis results for NGP and two alternative GP models where the observations are generated from a NGP prior.

using a tempered stable subordinator. Afterwards, in order to demonstrate the generalisation of a NGP to multidimensional input spaces a two dimensional example problem is presented using a toy data set available in TensorFlow (Wickham (2016)) on diamond prices.

Any stationary kernel function can be used to define the conditional kernel function in NGP regression. Here a squared exponential (SE) kernel is selected for the conditional GP. In order to demonstrate the ability of a NGP in identifying local characteristics, a length-scale $l = 0.1$ is used and an observation set defined on a small region of the input space $(0, 0.5)$ is simulated from a NGP prior. The latent transformation space in this example is generated as a TS subordinator with $\alpha = 0.8$ and $\beta = 5$ and the observations have i.i.d. noise with standard deviation $0.1$. The observations and results are shown in Fig. (1).

The NGP predictive posterior density shown in the first column of Fig. (1) clearly identifies some local changes in the variance that match with large jumps observed in the latent transformation posterior samples above. Furthermore, for input differences on the order of $l$ the density retains close to Gaussian behaviour. The posterior sample paths on the latent transformation space identify how fast the SE kernel decays to zero around different regions in $\mathcal{X}$. Large jump sizes break the correlation between local points and the associated observations are treated as statistically independent. From this perspective, if the model correctly identifies large jumps this shows that it discovers some observations contain more information about their local region than a stationary GP can encode. Hence, the sample paths provide an expressive probabilistic layer for interpreting non-Gaussian behaviour.

Two alternative GP regression results are presented in the second column of Fig. (1). Firstly, using the ground truth value of $l = 0.1$, a smooth approximation of the predictive posterior density can be obtained. Alternatively, a GP with a higher marginal likelihood can be obtained by optimising the length-scale. In order to adapt to large deviations in some regions of $\mathcal{X}$ the optimisation results in a smaller value of $l = 0.01$. This can be considered as a trade-off between in-sample performance and the generalisation capability of the model. As $l$ gets smaller each observation is modelled as almost statistically independent and there are no long distance dependencies between inputs. Such a representation will likely be an overfitted model that does not have any generalisation capability and interpolations will produce white-noise. NGP regression provides a more sparse representation of the random function in the sense that it only defines local statistical independence assumptions if the observations show non-Gaussian behaviour and otherwise retain long distance dependencies.

For the multidimensional case an example regression model for diamond prices is presented. The features used in this task are the carat of a diamond which is a measure of its weight and the percentage length of its table which is the largest flat facet of the diamond and affects how the diamond interacts with light. For ease of visualisation, both input dimensions are linearly transformed to lie between $[0, 1]$. The experiment is designed such that a 1000 randomly selected input-output pairs

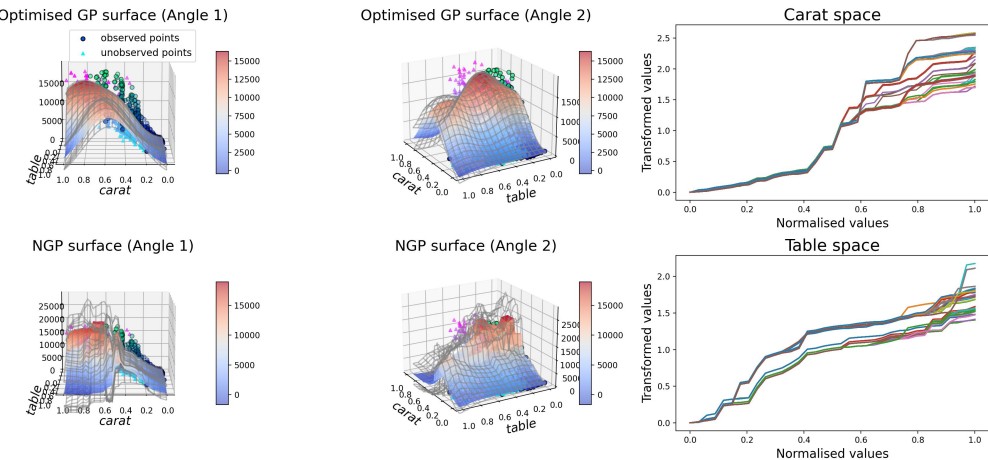

Figure 2: Regression analysis results for NGP and GP models with for the diamond price data set using a TS subordinator. The posterior means are plotted as a surface and the $\pm 3$ standard deviation surface is overlaid on the mean as a wireframe plot. The right hand column shows the posterior subordinator samples for a TS subordinator.

are chosen as the training set and the learned posterior surface is compared against another randomly selected 1000 pairs.

Figure (2) show the results of NGP regression using a TS subordinator. The main distinguishing property of the diamond price data set is that the price increases non-linearly with increases in the carat feature. The GP surface shown here forms a smooth function that cannot model the rapid change in prices around these regions of non-linear increase. Furthermore, the predictive surface is making non-zero predictions around out-of-sample regions which are undesirable in downstream decision-making tasks. The NGP surface is able to identify the non-linear increase in prices and the increased uncertainty in prices for larger carat values. These properties can be most clearly identified in the posterior subordinator samples. Note that the mean log conditional likelihood of the MCMC samples are found as $-97873.9$ and the GP log likelihood is found to be $-113731.8$.

NGP regression with a tempered stable subordinator presented in this work may be applied to datasets where there are local deviations from Gaussian behaviour but the overall trend of the function closely follows a GP. Using alternative characterisations of the subordinator, varying degrees of non-Gaussian behaviour can be modelled in a NGP regression framework as briefly demonstrated in Section 4. Some practical examples of subordinator processes are gamma processes (Rosiński (2001)) and inverse Gaussian processes (Rydberg (1997); Barndorff-Nielsen (1997)) which both lead to analytical probability density functions unlike the TS process. This fact can be utilised to design better proposal densities for a MCMC procedure. However given any Lévy density a similar formulation to Section 3.1 can be readily designed and used for inference as studied in Section 3.2. A particularly interesting case is the generalised inverse-Gaussian process which can capture various degrees of semi-heavy-tailed behaviour, including the gamma and inverse Gaussian processes (Eberlein & v. Hammerstein (2004); Godsill & Kındap (2021)).

Using NGP regression produces a generative model conditioned on a dataset where samples from both the posterior density over the input transformation and the predictive density over the observations can be generated. A probabilistic representation of a latent layer between the input and output spaces may lead to new insights about the underlying data generating mechanism. Furthermore, our construction of a NGP using the time-change operation may potentially be extended to any probabilistic setting for interpreting non-Gaussian behaviour. For example, Lévy fields can be used to model the first layer of a deep GP architecture (Titsias & Lawrence (2010); Damianou & Lawrence (2013)).

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

# A   APPENDIX

## A.1   ADDITIONAL DETAILS OF THE EXPERIMENTAL SETUP

For the first experimental setup, our Gibbs sampler defines a grid of 100 disjoint intervals and iterates over the whole space 50 times. After initial samples are discarded for burn-in the average log conditional likelihood of the remaining samples is found to be $51.46$ with a standard deviation of $3.17$. In comparison the log conditional likelihood of the data generating process is found as $59.09$ which suggests that the NGP does not overfit. The log marginal likelihood of the GPs with $l = 0.1$ and $0.01$ are found as $-388.39$ and $-71.38$, respectively. The mean acceptance probability for each step in Alg. 3 is found as $0.72$. The confidence intervals in regression results show 3 standard deviations. Lastly, on the right column we show 5 samples from the NGP posterior density and its smooth approximation.

Given the results above the two main aspects that require further attention are the design of a sensible prior distribution on the latent transformation space and the initialisation of the Gibbs sampler. The tempered stable (TS) subordinator is characterised by three parameters, $\alpha$, $\beta$ and $C$, that represent the tail heaviness, tempering and scale. The expected value and variance of the subordinator process on an input space $\mathcal{X}$ is a function of these parameters and the length (or measure) of $\mathcal{X}$. In order to produce results that are comparable with Gaussian process (GP) regression, the expected value of the subordinator process is set to the length of $\mathcal{X}$, i.e. given an interval $(x_{min}, x_{max})$ the expected value is $|x_{max} - x_{min}|$. As discussed in Section 2.2, if the observed input points are assumed to lie on a Euclidean space, the change in the covariance scales linearly according to $|x_{max} - x_{min}|$. On the latent transformation space, this corresponds to an identity map in $(x_{min}, x_{max})$. Setting the expected value to the length of the input set expresses a preference towards regular Gaussian behaviour and the deviations from the identity map provide insight into the characteristics of the observed data set.

Note again that the shot-noise simulation methods studied in Section 3.1 produce approximate sample paths from a Lévy process since the infinite series described by Eq. (3) have to be truncated to a finite number of terms. The convergence of the series is found in practice to be faster for smaller values of $\alpha$ and $\beta$ parameters. The number of terms that are required to obtain a sufficiently close estimate can be adaptively found using probabilistic asymptotic bounds and will be presented in future extensions of our work. However, a simple way to ensure the required convergence is to increase the number of terms produced as $\beta$ gets larger. In practice for $\beta = 5$, producing 1000 terms in Alg. 1 is found to work well.

The initialisation of the Gibbs sampler in practice is one of the central issues in designing Markov chain Monte Carlo methods. In Alg. 3, the sampling method depends on the number and size of small disjoint intervals $\tau$. As the size of $\tau$ is decreased, the acceptance rates for the Gibbs

sampler increase and convergence can be achieved in a few number of iterations of the whole input space. However, this also results in an increased number of intervals and each iteration requires more computation and time. Our strategy to initialise the Gibbs sampler is to obtain a crude estimate of the latent transformation function by running the same algorithm on a few number of intervals $\tau$ and set the initial state of the Gibbs sampler to the jump magnitudes and positions that correspond to the last sample path. The chain can then be simulated starting from this state and only a few samples have to be discarded as burn-in. Alternatively, a simpler initialisation is to generate linearly spaced points with equal magnitudes that represent the identity map corresponding to a GP regression setting. This alternative initialisation can be particularly useful for dataset that display close to Gaussian behaviour.

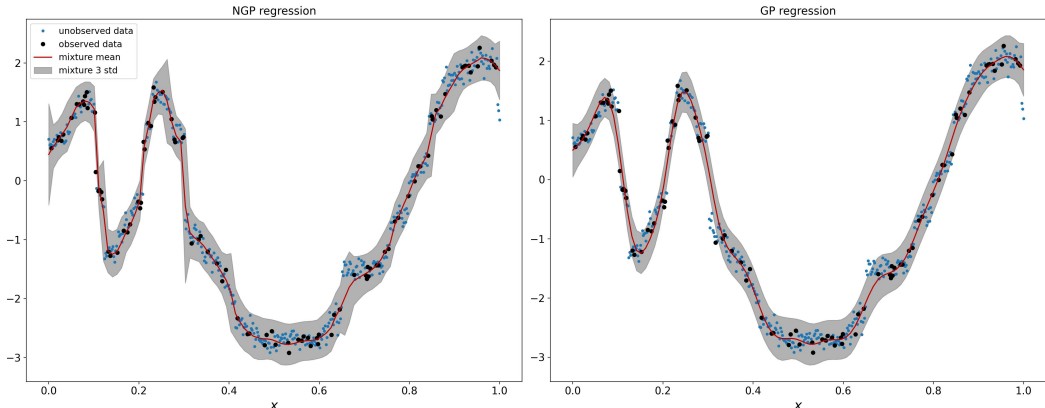

Figure 3: Regression analysis results for NGP and GP models with the Matérn kernel where the observations are generated from a NGP prior.

It is straightforward to change the particular choice of a kernel function without any changes in our presented algorithms. In Fig. 3 an example regression problem is presented using the conditional kernel function specified as the Matérn kernel. The Matérn kernel decays slower than a squared exponential kernel and therefore can be used to model long range dependencies. Particular parameter values of the Matérn kernel correspond to the covariance of certain stochastic differential equations as described in Whittle (1963). The results are again compared with a GP regression setting to highlight the differences. The smoothness parameter is chosen as $\nu = 5/2$ and the length-scale is $l = 0.1$. The subordinator parameters are identical to the previous example in Fig. 1. A dataset of size $500$ is generated on $(0, 1)$ from the NGP prior and a set of size $100$ is randomly selected as the observations. In this case, the unobserved elements in the dataset serve as the test set. For the NGP regression, most unobserved data points lie inside the confidence intervals while the GP regression misses these points around local deviations. The log conditional likelihood of the NGP and the marginal likelihood of the GP models are $18.25$ and $-114.97$ respectively.

Additionally, in order to emphasise that our framework works for any choice of Lévy process with a measure that possesses a density function, the multidimensional experiment is run separately using a gamma subordinator. The required simulation algorithm for the gamma subordinator is presented in Rosiński (2001); Godsill & Kındap (2021). The training set is independently sampled and the results are presented Figures 4 and 5. Similar conclusions can be made using the gamma subordinator. However note that the gamma subordinator is characterised by the flat intervals shown in Fig. 5 and there may be other applications that are more appropriate for such a posterior surface.

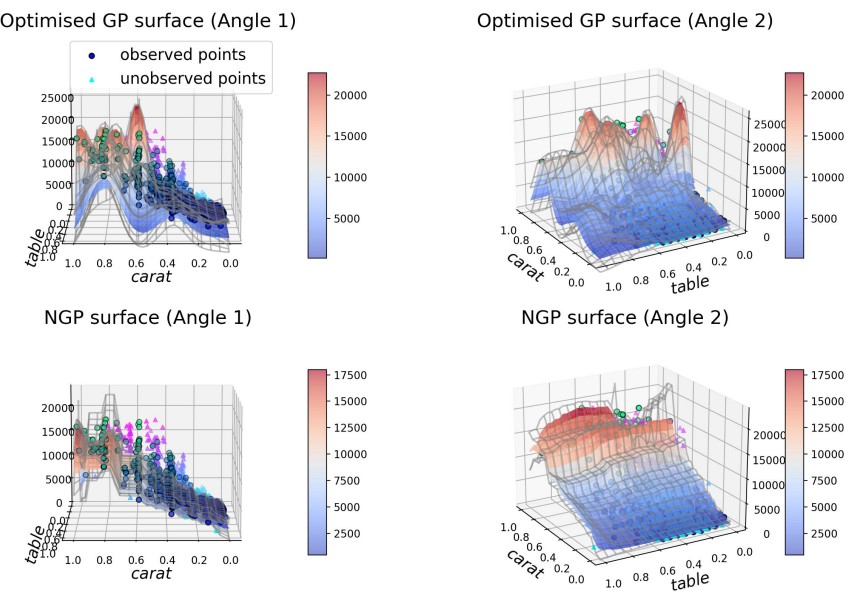

Figure 4: Regression analysis results for NGP and GP models with for the diamond price data set using a gamma subordinator. The posterior means are plotted as a surface and the $\pm 3$ standard deviation surface is overlaid on the mean as a wireframe plot.

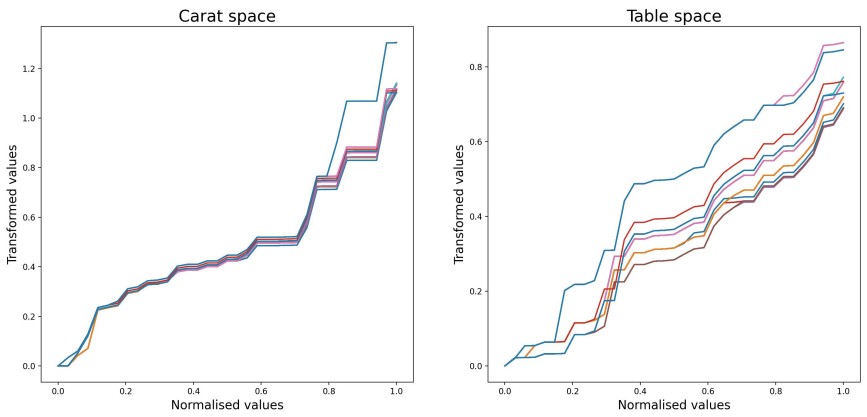

Figure 5: Posterior subordinator samples for a gamma subordinator.

