# OpenReview forum: "Non-Gaussian Process Regression"
_ICLR.cc/2023/Conference — Submitted to ICLR 2023_

### Official Review · Reviewer_AecU · 2022-10-14

**Confidence:** 4
**Correctness:** 3
**Technical Novelty And Significance:** 2
**Empirical Novelty And Significance:** 2
**Recommendation:** 5

**Clarity, Quality, Novelty And Reproducibility:**

The description of the proposed model and its posterior simulation are clear; the novelty seems low relative to existing methods such as treed GPs; reproducibility is satisfactory.

**Strength And Weaknesses:**

The construction of the model and the strategy described for its computational implementation (i.e., approximate posterior simulation) appear to be sound and well thought out.  Yet the motivation for developing this type of model and its inferential strengths and weaknesses over other approaches are not established to my satisfaction.

It is certainly the case that non-smoothness is evident in many real world datasets, and at times this may be such a magnitude that the smooth functional form preferred (or even 'demanded', according to the nature of the kernel) under the Gaussian process prior becomes a hinderance to effective inference.  The consequences of this model misspecification and the pathway to appropriate model revisions is highly context dependant.  In a general sense this means that the objectives and conditions of the inference must be specified clearly---ideally through an appropriate utility function---and the nature of the robustness sought must be established. E.g. is the exercise one of smoothing over noisy observations along a well-sampled but noisily observed window of data with the objective of minimising mean squared error relative to the true latent function, or is the focus rather on interpolation over a missing segment of the curve with the utility being a Frequentist coverage performance of the pointwise posterior bounds?  Only once these aims are established can one then effectively investigate the performance of the proposed solution, both in absolute terms and relative to competing methods.  This investigation can reasonably be either via computational experiments as in the present manuscript or via theoretical study; with ideally the two aspects of the literature complementing each other.  In the former (computational experiments) strength is given by the demonstration of substantive computational and/or accuracy superiority over existing methods in real world problems, or by demonstration on mock datasets that the new method solves a meaningful failure mode of the existing solution.

In the present set of experiments it seems that the exercise is one of smoothing noisy data (more so than interpolation) with discontinuities such a those drawn from the proposed NGP process itself, and the utility lies in the reduced degree of oversmoothing that results compared with the base Gaussian process.  In this setting it would seem that the natural comparison points are the likes of Gramacy & Lee's Bayesian treed GP model (JASA 2008) and Kim, Mallick & Holmes' piecewise Gaussian processes (JASA 2005).  Models in which the mean function of the GP is treated as a piecewise linear process would also be a natural alternative: why does the discontinuity need to be put through the kernel?  Indeed, one also thinks of Frequentist non-parametric regression methods (e.g. piecewise linear regression) here too: what is the Bayesian aspect of this problem adding from an inferential point of view (because it's unlikely to be competitive in speed with the need to do full posterior sampling)?  How is the hyper-prior specification complicated by the 'degeneracy' between the Levy process parameterisation and the kernel parameterisation?


**Summary Of The Paper:**

In the manuscript, "Non-Gaussian Process Regression", the authors present a Bayesian modelling strategy for semi-parametric regression in which the standard setting of Gaussian process regression is augmented with a transformation of the input space through a 'time change' operation.  In particular, the latter is based on a Levy process to allow the regression function to better follow apparent discontinuities than can the original Gaussian process.

**Summary Of The Review:**

Overall, I think the authors have the start of a useful piece of work here (namely, the formulation and implementation of an interesting model), but in the present exposition I do not find a compelling case for becoming interested in this model over any other type of 'fancy' Gaussian process.

---

> ### Author Response · Authors · 2022-11-18
> **Clarifications on identified weaknesses**
>
> We would like to thank the reviewer for their helpful comments.
>
> Specifically, the example experiments shown in the paper will be updated to clarify the relevant decision-making problems under consideration instead of just focussing on the posterior density fit. Furthermore, additional comparisons of the NGP model against alternative extended GP formulations will be added to the paper. Thank you for providing additional literature on relevant research.
>
> Our future works will present hyper-parameter learning methods incorporated within the sampling procedure which will clarify how the kernel parameters and subordinator parameters are related.
>
> While it is certainly true that there may be frequentist non-parametric inference methods that have less time complexity, the ability to obtain a probabilistic representation of each layer in the model is required by many decision-making problems with high-impact potential such as design of drugs, financial modelling, etc.
>
> The NGP model can be scaled to higher dimensions efficiently by considering a lower dimensional subordinator space compared to the input space where each dimension is transformed by a linear combination of the subordinators. The dimensions of the subordinator process is effectively another hyper-parameter which increases the complexity of the model as it increases. These discussions will be added to future versions of this work.

---

### Official Review · Reviewer_bFEN · 2022-10-24

**Confidence:** 3
**Correctness:** 3
**Technical Novelty And Significance:** 3
**Empirical Novelty And Significance:** 2
**Recommendation:** 5

**Clarity, Quality, Novelty And Reproducibility:**

The work is novel in the sense of using a latent transformed input space and the Levy process for modelling the random evolution of the latent transformation. The proposed approach is potentially promising, but it needs a more clear motivation with stronger examples (or at least a more extensive explanation with respect to the current examples). The sampling method is rather an application of an existing sampler to this particular problem.

**Strength And Weaknesses:**

Strengths:
-- The proposed method is interesting and promising for cases when one wants to use a GP, but the data are possibly non-Gaussian with non-stationary jumps.
-- The paper is overall clear and well presented.

Weaknesses:
-- A more extensive comparison with possible competitors is missing.
-- The examples demonstrating the performance of the method are quite limited. In particular, it would be nice to see examples with longer data sets containing more jumps.
-- The limitations of the method (computational and conceptual are not completely clear; how many dimensions are feasible; how many jumps are identifiable; how robust is the method to initialization of the Gibbs sampler etc.).

Detailed comments:
•	Is the MCMC algorithm feasible in higher dimensions? Can sample paths in different dimensions be dependent?
•	There are only 2 points with jumps in Figure 1; what would happen with a longer data series with more jumps (say above 100)? Is that still feasible to identify them? Would the inference be possible in this case, and what effect would initialization in Gibbs sampler have here?
•	If the interest is in the location of the jumps, what would be the advantage of NGP compared to a GP regression and, say, change point kernel with Matern as base kernels?
•	If we assume some non-stationarity, how the fitted results would be different between NGP and GP with a Matern kernel? What would be the advantage of using NGP over a standard GP regression with the Matern kernel? I further noticed that this kind of experiment is presented in Appendix A1, but the advantages are still not completely clear. Why NGP fit is considered to be better?
•	It would be interesting to see how NGP performs out of sample, i.e. what happens with the predictions if you extrapolate GP to the right in Figures 1 and 3.



**Summary Of The Paper:**

The paper proposed a non-Gaussian process regression which allows to model GPs with time changes and thus accounts for non-Gaussian behaviours such as heavy tails. The model is constructed using a latent transformed input space, and the Levy process is used to model the random evolution of the latent transformation. Further, MCMC (MH-within-Gibbs sampler) is used for the inference.

**Summary Of The Review:**

The paper proposes an interesting method for modelling potentially non-Gaussian data with heavy tales. However, the current evaluation is rather limited and does not fully demonstrate the usefulness of the method compared to standard GP regression with Matern kernel and GP regression with change point kernel with Matern kernels as base kernels. From the presented examples, some limitations of the method also cannot be identified, i.e. how many jumps are feasible in terms of the inference, how robust MCMC will be when there are more jumps in the process (for example, does initializing Gibbs sampler become harder?).

---

> ### Author Response · Authors · 2022-11-18
> **Clarifications on identified weaknesses**
>
> We would like to thank the reviewer for their helpful comments.
>
> Firstly, our experiments suggest that the Gibbs sampler is robust to the initialisation as the design allows the granularity of the Gibbs marginals to be dynamically set. The number of significantly large jumps can be identified by adjusting the number of disjoint intervals being considered in the sampler and a choice of a small number of intervals result in rapid convergence. The fit of the posterior distribution can be further optimised depending on downstream needs by dynamically increasing the number of intervals which leads to smaller adjustments at each iteration with a higher acceptance probability.
>
> Additionally, we consider Levy processes of infinite-activity, meaning that there are an infinite number of jumps in any interval. Thus, it is impossible to identify all jumps but a probabilistic argument can be made for jump sizes greater than some constant epsilon. In future versions of this work, we'll present these results which are also central to the associated simulation algorithms.
>
> In terms of scaling the model, the dimensions of the latent transformation space may differ from the input dimensions such that the transformation corresponding to a particular dimension may be a linear combination of a smaller set of subordinator processes. This also implies that it is possible to encode dependence between separate input dimensions. However, in future work we'll provide an exact analysis of the time complexity of our Gibbs sampler
>
> When comparing the NGP with some particular GP, we would like to focus on the overall generative model of the data set and the representation of uncertainty instead of specific loss metrics as these are prone to overfitted results depending on the given data set. Hence the GP models presented in the paper are kept as simple as possible to be interpretable. Our goal then is to show that these simple definitions can be made greatly more flexible through the additional time-change operation while still offering an interpretable description of the data generating system.
>
> The learned transformation function in the NGP also defines a random transformation anywhere in the input space since Levy processes are stationary. Thus if we can correctly identify the characteristics of the posterior subordinator function, we also learn an additional layer of variability. Whereas a GP can only learn information about a particular subset of the input space where it has seen observations, the NGP allows some potential information about the unobserved regions of the input space to be gleaned. This is the primary reason NGP’s are considered more powerful than GP representations.
>
> We agree that having multiple GP specifications using change-point kernels and Matern kernels to compare against the NGP results will be useful and in future work we’ll consider additional experiments to clarify the use of NGPs.

---

### Official Review · Reviewer_ebws · 2022-10-25

**Confidence:** 4
**Correctness:** 4
**Technical Novelty And Significance:** 3
**Empirical Novelty And Significance:** 2
**Recommendation:** 3

**Clarity, Quality, Novelty And Reproducibility:**

The paper is clear and the derived sampler is novel. The theoretical derivations are good in quality.

**Strength And Weaknesses:**

Strengths:
- Clear and well written
- Technically correct
- Effort in the development of an efficient, non-trivial MCMC

Weaknesses:

- Only tested in 1D and 2D regression
- The hyperparameters of the NGP aren't tuned with this procedure
- The 1D problem is a toy example in which the data follows exactly the model
In the 2D problem it is unknown what MSE is, and how it is affected by the choice of hyperparameters. The standard GP seems oversmoothed. If the lengthscales are tuned, is it not possible to get a similar MSE to that of the NGP? This very fundamental question is left unanswered. Given the dimensionality of the data, both lengthscales could be swept to properly check this conclusively. It says "optimized GP", so I'm assuming that some lengthscale selection is happening, but his might be falling in a local minimum. The surface is clearly oversmoothed. Also, other kernels should be tested to attempt best match to the data while still remaining within the tractability of the GP.

**Summary Of The Paper:**

Mechanisms for MCMC inference in NGP are introduced. NGPs are defined GPs in which the input has been warped according to a monotonically non-decreasing function with a pure jump Lévy process prior.

**Summary Of The Review:**

An interesting theoretical derivation, but this paper looks more like a solution in search of a problem. The experiments are unconvincing for being in lower dimension and not try hard enough to make the standard GP look good (measuring the MSE for the best hyperparameter selection -- instead measuring only the log-likelihood for what looks like an inadequate lengthscale/kernel selection).

---

> ### Author Response · Authors · 2022-11-18
> **Clarifications on identified weaknesses**
>
> We would like to thank the reviewer for their helpful comments.
>
> The experimental results are presented for both 1d and 2d cases to emphasise that the formulation is valid for an arbitrary number of dimensions. Specifically, the dimensions of the latent transformation space may differ from the input dimensions such that the transformation corresponding to a particular dimension may be a linear combination of a smaller set of subordinator processes.
>
> We'll present hyperparameter tuning within the procedure in a future work.
>
> When comparing the NGP with some particular GP, we would like to focus on the overall generative model of the data set and the representation of uncertainty instead of specific loss metrics as these are prone to overfitted results depending on the given data set. Hence the GP models presented in the paper are kept as simple as possible to be interpretable and tuned using standard Python libraries. Our goal then is to show that these simple definitions can be made greatly more flexible through the additional time-change operation while still offering an interpretable description of the data generating system. However, we agree that having multiple GP specifications to compare against the NGP results will be useful. In future work, we’ll consider additional experiments to clarify the use of NGPs.

---

### Decision · Program_Chairs · 2023-01-20

**Decision:**

Reject

**Justification For Why Not Higher Score:**

Reviewers were in agreement

**Justification For Why Not Lower Score:**

N/A

**Metareview: Summary, Strengths And Weaknesses:**

The authors present a new time-changed GP model which they say is suitable for the modeling of heavy-tailed non-Gaussian processes. While there is considerable theoretical and practical development in the present manuscript, the reviewers found the experimental section ultimately unconvincing. As the authors agreed, there are a number of future experiments and comparisons that would significantly strengthen the work: change-point kernels, hyper-parameter learning, extensions to higher-dimensionality, and so on. I agree with with the reviewer who suggested that a comparison to treed GPs would be desirable. I would also question the authors' use of the term "interpretable" and suggest their claims of interpretability should be further explored in an updated manuscript.